# Effects of Total Thermal Balance on the Thermal Energy Absorbed or Released by a High-Temperature Phase Change Material

**DOI:** 10.3390/molecules26020365

**Published:** 2021-01-12

**Authors:** Suset Rodríguez-Alemán, Ernesto M. Hernández-Cooper, Rolando Pérez-Álvarez, José A. Otero

**Affiliations:** 1Tecnológico de Monterrey, Escuela de Ingeniería y Ciencias, Carr. al Lago de Guadalupe Km. 3.5, State of Mexico 52926, Mexico; susetuh@gmail.com (S.R.-A.); emcooper@tec.mx (E.M.H.-C.); 2Centro de Investigación en Ciencias, Universidad Autónoma del Estado de Morelos, Cuernavaca, Morelos 52900, Mexico; rpa@uaem.mx

**Keywords:** total thermal balance, high-temperature phase change material, thermal energy storage

## Abstract

Front tracking and enthalpy methods used to study phase change processes are based on a local thermal energy balance at the liquid–solid interface where mass accommodation methods are also used to account for the density change during the phase transition. Recently, it has been shown that a local thermal balance at the interface does not reproduce the thermodynamic equilibrium in adiabatic systems. Total thermal balance through the entire liquid–solid system can predict the correct thermodynamic equilibrium values of melted (solidified) mass, system size, and interface position. In this work, total thermal balance is applied to systems with isothermal–adiabatic boundary conditions to estimate the sensible and latent heat stored (released) by ${\mathrm{KNO}}_{3}$ and ${\mathrm{KNO}}_{3}/{\mathrm{NaNO}}_{3}$ salts which are used as high-temperature phase change materials. Relative percent differences between the solutions obtained with a local thermal balance at the interface and a total thermal balance for the thermal energy absorbed or released by high-temperature phase change materials are obtained. According to the total thermal balance proposed, a correction to the liquid–solid interface dynamics is introduced, which accounts for an extra amount of energy absorbed or released during the phase transition. It is shown that melting or solidification rates are modified by using a total thermal balance through the entire system. Finally, the numerical and semi-analytical methods illustrate that volume changes and the fraction of melted (solidified) solid (liquid) estimated through a local thermal balance at the interface are not invariant in adiabatic systems. The invariance of numerical and semi-analytical solutions in adiabatic systems is significantly improved through the proposed model.

## 1. Introduction

High-temperature phase change materials (HTPCMs) represent an appealing application as latent heat and sensible heat storage devices used as backup systems for thermoelectric generation [1,2]. Concentrating solar power (CSP) plants use solar power to generate superheated steam during the hours of the day with the highest solar irradiance. Latent heat and sensible heat storage devices are widely used to increase the renewable energy efficiency of CSP plants by providing thermal energy during the hours of the day with unusable solar power. On the one hand, extensive research has been performed to improve mathematical models that can predict the energy density and charging/discharging times in several configurations of thermal energy storage (TES) units [3,4,5]. On the other hand, experimental work has been performed to study the effect of device geometries [6] and composite based heat storage units [7,8,9] in order to enhance the heat transfer rates between the PCM and the heat transfer fluid (HTF). The latent heat and melting temperature of salts used as HTPCMs have been determined through molecular dynamics and ab initio simulations [10]. Mass accommodation methods have been developed to include the effects of density variations produced by pressure increments during melting of encapsulated PCMs [4,11,12,13,14]. The pressure increments on spherically microencapsulated HTPCMs have been predicted through models that assume incompressible solid phases [4]. The latent heat energy density is diminished by pressure increments, and different models have been proposed to estimate the latent heat during melting of encapsulated PCMs [4,5,14]. The research has been focused on improving the desired thermodynamic properties of PCMs to enhance the heat energy transfer rates between the PCM and the HTF, and increase the thermal energy density of PCMs.

The phase change process related with PCMs lies within the moving boundary type of problems. Two phase one-dimensional problems have been extensively studied in the literature, and there are a few cases for which it is possible to find an exact analytical solution [15,16]. For this reason, different numerical and semi-analytical methods have been used to find approximate solutions to these type of problems [17,18]. The Finite Element Method (FEM) is one of the most popular numerical methods used to solve systems of differential equations. There is an extensive amount of research devoted to develop more accurate and efficient solutions to heat transfer related phenomena through the FEM, including the dynamics of first-order phase transitions [17]. Explicit FEMs have been proposed to find solutions for the temperature distribution in non-linear heat transfer problems, where the temperature dependence of the thermodynamic variables of specific heat and thermal conductivity were taken into consideration [19]. Modifications to the FEM have also been used to find more accurate solutions to the liquid–solid phase transition [17,20,21].

The non-linearity introduced by the thermal balance at the interface during a first-order phase transition has also been studied through other numerical and semi-analytical methods. Explicit and implicit finite difference methods (FDMs) have also been used to find solutions for the one-dimensional liquid–solid phase transition at constant pressure [18,22,23]. The advantages of FEMs over FDMs rely on the way in which adiabatic boundary conditions are implemented and the interpolation methods needed to calculate the temperature in front tracking problems. Adiabatic systems have been used to find corrections to mathematical models of phase change processes, where total energy conservation plays a key role in these type of systems [23]. On the one hand, adiabatic boundary conditions do not need to be approximated when using FEMs, allowing well-behaved solutions at thermodynamic equilibrium. On the other hand, high-order approximations to the spatial derivative involved in this type of boundary conditions are needed when using FDMs, in order to reach the expected state of the system at thermodynamic equilibrium [23]. Semi-analytical approaches have also been used to find solutions for the liquid–solid interface dynamics through the heat balance integral method (HBIM) [23,24] and the refined heat balance integral method (RHBIM) [13,14,25]. Finally, the accuracy of the proposed solutions to the moving boundary problem when the system is subjected to different types of boundary conditions has been tested with the available similarity solutions, other numerical solutions, or experimental results [26,27,28].

This work is concerned with some of the fundamental aspects during liquid–solid isobaric phase transitions in one-dimensional configurations. The phase change process is held at constant pressure (isobaric phase transition), where volume changes produced by the density difference between liquid and solid phases are taken into consideration by imposing total mass conservation of the liquid–solid sample. The goal is to take a further step into the estimation of the energy capacity of HTPCMs and charging/discharging times. Recently, higher-order corrections to the liquid–solid interface dynamics have been introduced by proposing a total thermal balance through the entire system instead of the more classical local thermal balance at the interface [29]. These corrections are not intuitive and could be interpreted through an equivalent latent heat of fusion using the Leibniz integral rule. In this work, the effects of total thermal balance in the energy capacity of HTPCMs and charging/discharging times are estimated. The precision of the numerical and semi-analytical methods is relevant for estimating the higher-order contributions from total thermal balance in the thermal energy absorbed or released by a PCM sample. The RHBIM and FEM are used to find solutions to the proposed and classical models. The analysis of the thermal energy absorbed/released by the PCM indicates that the equivalent latent incorporates an extra or missing thermal energy contribution during the phase transition when compared with the predictions obtained from the classical model. The equivalent latent heat can be pictured as an apparent latent heat, since it introduces corrections to the thermal energy absorption/release rates, but the latent heat absorbed is shown to be related only with the bulk latent heat of fusion. The consistency of total thermal balance is determined through the invariance of the numerical and semi-analytical solutions in adiabatic systems, whether the volume changes of the sample are incorporated by assuming a right or left moving boundary. Additionally, numerical and semi-analytical solutions to invariant quantities such as volume changes and the fraction of melted (solidified) solid (liquid) are found to reach the same thermodynamic equilibrium values in adiabatic (thermally isolated) systems. Finally, the contributions from total thermal balance are determined through the numerical and semi-analytical solutions to the total energy absorbed (released) during a charging (discharging) process in ${\mathrm{KNO}}_{3}$ and ${\mathrm{KNO}}_{3}/{\mathrm{NaNO}}_{3}$ salts.

## 2. Model and Methods

### 2.1. Description of the Physical System

Consider a sample of size L(t) and cross-section *A* with a liquid phase in contact with a solid phase, separated by an interface with position ξ(t) at some melting temperature $T_{m}$. The sample has a left (right) boundary at $x_{\ell }\left(t\right)$$\left(x_{s}\left(t\right)\right)$, which obeys an equation of motion that imposes total mass conservation. The net flux of thermal energy through the interface causes its displacement in time *t*. The volume changes during the phase transition are incorporated through the dynamic variable $x_{\ell }\left(t\right)$$\left(x_{s}\left(t\right)\right)$. The heat transfer through the system is homogeneous about the perpendicular plane to the heat flux. The temperature of the sample at some position *x* along the longitudinal axis and at some time *t* is considered to be uniform along the y−z plane. The thermal flux only takes place along the longitudinal direction *x* since there are no temperature gradients along the y−z plane. Under this consideration, the problem becomes one-dimensional, and the interface dynamics will only take place along the longitudinal *x*-direction. The thermodynamic variables of thermal conductivity and specific heat capacity are assumed to be independent of the temperature. Thermal expansion through temperature-dependent densities is also neglected; then, natural convection driven by buoyancy is not considered. The shrinkage or expansion of the sample, due to solidification or melting, is held at constant pressure. Natural convection induced by pressure gradients within the liquid phase can be neglected. The liquid (ℓ) and solid (*s*) phases have a temperature profile $T_{i}\left(x,t\right)$, where i=ℓ,s. Here, the temperature at any point within the liquid (solid) is above (below) the melting temperature $T_{m}$, which is constant in time and does not depend on the spatial variable *x*.

### 2.2. Adiabatic Systems

The system is thermally isolated; therefore, it will be subjected to the following adiabatic boundary conditions:(1)$$\frac{\partial T_{\ell }\left(x,t\right)}{\partial x}{|}_{x=x_{\ell }\left(t\right)}=0,\;\frac{\partial T_{s}\left(x,t\right)}{\partial x}{|}_{x=x_{s}\left(t\right)}=0,$$
and an isothermal boundary condition at the interface
(2)$$T_{\ell }\left(\xi \left(t\right),t\right)=T_{s}\left(\xi \left(t\right),t\right)=T_{m},$$
where ξ(t) represents the position of the liquid–solid interface at some time *t*. Equation (1) means that the system is thermally isolated from the surroundings, which implies that thermal energy is only transferred between the liquid and solid phases. The boundary condition at the liquid–solid interface ξ(t), given by Equation (2), is used to describe the isothermal nature of the phase transition. The melting or solidification process will take place at a constant temperature equal to the melting temperature $T_{m}$ of the PCM. The initial temperature profiles will be assumed to be polynomial functions of *x*:(3)$$T_{i}\left(x,0\right)=f_{i}\left(x\right),\;\mathrm{with}\;i=\ell ,s,$$
where $f_{i}\left(x\right)$ is a polynomial function that represents the initial temperature distribution at phase *i*. The initial temperature profile can be obtained in order to satisfy the boundary conditions given by Equations (1) and (2). Initially, there is a certain amount of liquid and solid in the sample so that ξ(0)>0.

Since the thermodynamic variables are independent of the temperature, the heat equation in each medium can be written as follows:(4)$$\frac{\partial T_{i}\left(x,t\right)}{\partial t}=\alpha_{i}\frac{\partial^{2}T_{i}\left(x,t\right)}{\partial x^{2}},$$
where $\alpha_{i}=k_{i}/\left(\rho_{i}\;C_{i}\right)$ is the heat diffusion coefficient of phase *i*, and is defined in terms of the thermal conductivity $k_{i}$, specific heat $C_{i}$ and density $\rho_{i}$.

The thermal energy balance at the liquid–solid interface ξ(t) that is consistent with mass conservation is given by the following equations:
(5a)$$L_{f}\;\rho_{\ell }\frac{d\xi (t)}{dt}=-k_{\ell }\;\frac{\partial \;T_{\ell }\left(x,t\right)}{\partial \;x}{|}_{x=\xi (t)}+k_{s}\;\frac{\partial \;T_{s}\left(x,t\right)}{\partial \;x}{|}_{x=\xi (t)}\;\;\;\;,\mathrm{or}$$
(5b)$$L_{f}\;\rho_{s}\frac{d\xi (t)}{dt}=-k_{\ell }\;\frac{\partial \;T_{\ell }\left(x,t\right)}{\partial \;x}{|}_{x=\xi (t)}+k_{s}\;\frac{\partial \;T_{s}\left(x,t\right)}{\partial \;x}{|}_{x=\xi (t)}\;.$$
where $L_{f}=\left(C_{\ell }-C_{s}\right)T_{m}$ is the bulk latent heat of fusion, and $C_{\ell }$ ($C_{s}$) is the specific heat capacity of the liquid (solid) close to the melting temperature $T_{m}$. On the one hand, Equation (5a) is the equation of motion for the interface when $x_{s}\left(t\right)$ is chosen as a moving boundary and $x_{\ell }\left(t\right)$ is constant in time. On the other hand, Equation (5b) is the equation of motion for ξ(t), when $x_{\ell }\left(t\right)$ is chosen as a moving boundary and $x_{s}\left(t\right)$ is constant in time, as shown in Ref. [23]. Equations (5a) and (5b) result from a local energy balance at the the liquid–solid interface.

The volume change experienced by the system due to the melting (solidification) of a small solid (liquid) slab is conceived by imposing total mass conservation of the system. Assuming that heat transfer is homogeneous in the direction perpendicular to the thermal flux, mass conservation can be imposed through the following equations of motion [23]:
(6a)$$\rho_{\ell }\frac{d\xi (t)}{dt}+\rho_{s}\left(\frac{d\;x_{s}\left(t\right)}{dt}-\frac{d\xi (t)}{dt}\right)=0,\mathrm{or}$$
(6b)$$\rho_{\ell }\left(\frac{d\xi (t)}{dt}-\frac{d\;x_{\ell }\left(t\right)}{dt}\right)-\rho_{s}\frac{d\xi (t)}{dt}=0\;.$$

Equations (5a) and (6a) are used to estimate the amount of melted (solidified) solid (liquid) and the total volume change of the sample when $x_{s}\left(t\right)$ is chosen as the dynamical variable. Equations (5b) and (6b) are solved when $x_{\ell }\left(t\right)$ is the dynamical variable [23]. The equation of motion for the left (right) boundary, coupled to the equation of motion for the liquid–solid interface position, has been able to reproduce the correct thermodynamic equilibrium values in some cases. Recently, it has been shown that the local thermal balance at the interface cannot predict thermodynamic equilibrium values for $\xi_{\mathrm{eq}}$, system size, and melted (solidified) mass in high-temperature materials [29]. Total thermal balance was proposed by forcing energy conservation in adiabatic systems. The energy and mass balance proposed resulted in a higher-order contribution to the equation of motion for the interface as follows:
(7a)$$\begin{array}{ccc} \rho_{\ell }\left(L_{f}-\left(\rho_{s}/\rho_{\ell }-1\right)\;C_{s}\left(T_{m}-T_{s}\left(x,t\right){|}_{x=x_{s}\left(t\right)}\right)\right)\frac{d\xi (t)}{dt} & = & -k_{\ell }\;\frac{\partial \;T_{\ell }\left(x,t\right)}{\partial \;x}{|}_{x=\xi (t)}\\ & & +k_{s}\;\frac{\partial \;T_{s}\left(x,t\right)}{\partial \;x}{|}_{x=\xi (t)},\mathrm{or} \end{array}$$
(7b)$$\begin{array}{ccc} \rho_{s}\left(L_{f}+\left(1-\rho_{\ell }/\rho_{s}\right)\;C_{\ell }\left(T_{\ell }\left(x,t\right){|}_{x=x_{\ell }\left(t\right)}-T_{m}\right)\right)\frac{d\xi (t)}{dt} & = & -k_{\ell }\;\frac{\partial \;T_{\ell }\left(x,t\right)}{\partial \;x}{|}_{x=\xi (t)}\\ & & +k_{s}\;\frac{\partial \;T_{s}\left(x,t\right)}{\partial \;x}{|}_{x=\xi (t)}. \end{array}$$

Equation (7a) or Equation (7b) incorporate the corrections obtained from total thermal balance when $x_{s}\left(t\right)$ or $x_{\ell }\left(t\right)$ is chosen as the dynamical variable. The above equations of motion take into account an extra amount of thermal energy transfer during the phase change process. The higher-order contribution obtained by imposing total energy conservation in adiabatic systems was pictured through an apparent latent heat [29], given by
(8a)$$L_{f}^{*}=L_{f}-\left(\rho_{s}/\rho_{\ell }-1\right)\;C_{s}\left(T_{m}-T_{s}\left(x,t\right){|}_{x=x_{s}\left(t\right)}\right),\;\;\mathrm{or}$$
(8b)$$L_{f}^{*}=L_{f}+\left(1-\rho_{\ell }/\rho_{s}\right)\;C_{\ell }\left(T_{\ell }\left(x,t\right){|}_{x=x_{\ell }\left(t\right)}-T_{m}\right).$$

Using these definitions, the previous form of the equation of motion for ξ(t) shown through Equation (5a) or Equation (5b) is recovered. Total thermal balance contributions are incorporated in the apparent latent heat defined through Equation (8a) or Equation (8b). Total thermal balance contributions depend on the boundary conditions and relative density difference between both phases as shown through Equations (8a) and (8b). On the one hand, materials used for thermoelectric generation, such as salts with high melting temperatures, are subjected to wide temperature ranges, where the contributions from the apparent latent heat are significant. On the other hand, PCMs used in thermal shielding applications are subjected to operating temperatures close to the melting point of the material, and the effects of total thermal balance are practically negligible.

Thermodynamic equilibrium values for the volume change of the system and fraction of melted (solidified) solid (liquid) have been previously obtained [23]. These equilibrium values have been used to identify inconsistencies in the predictions of models proposed by other authors. The volume change and fraction of melted (solidified) solid (liquid) can be obtained through an energy balance between the liquid and solid. The energy balance allows to find the amount of energy ΔU that will be absorbed (released) by a mass of solid (liquid) $\Delta \;M_{s}$ ($\Delta \;M_{\ell }$) to produce the phase transition between an initial state and the stationary state. The energy used to melt (solidify) a given mass of solid (liquid) $\Delta \;M_{s}$ ($\Delta \;M_{\ell }$) is given by
(9)$$\begin{array}{c} \Delta \;U=\rho_{\ell }\;C_{\ell }\;\int_{x_{\ell }\left(0\right)}^{\xi (0)}(T_{\ell }\left(x,0\right)-T_{m})dx-\rho_{s}\;C_{s}\;\int_{\xi (0)}^{x_{s}\left(0\right)}(T_{m}-T_{s}\left(x,0\right))dx\;, \end{array}$$
where $T_{\ell }\left(x,0\right)$$\left(T_{s}\left(x,0\right)\right)$ is the initial temperature profile in the liquid (solid) domain. Thermodynamic equilibrium values for the volume change and fraction of melted (solidified) solid (liquid) [23] are given by
(10)$$\begin{array}{c} \Delta \;L_{\mathrm{eq}}=\left(\frac{1}{\rho_{\ell }}-\frac{1}{\rho_{s}}\right)\frac{\Delta \;U}{L_{f}},\;\;\;\;\mathrm{and} \end{array}$$
(11)$$\begin{array}{c} f_{i_{\mathrm{eq}}}=\frac{\Delta \;M_{{\mathrm{eq}}_{i}}}{M_{i}\left(0\right)}=\frac{\left|\Delta \;U\right|}{M_{i}\left(0\right)\;L_{f}}, \end{array}$$
where $f_{i}$ is the fraction of melted solid $f_{s_{\mathrm{eq}}}$ or solidified liquid $f_{\ell_{\mathrm{eq}}}$ at thermodynamic equilibrium, and $M_{i}\left(0\right)$ is the initial mass of phase *i*. According to energy conservation in adiabatic systems, the values at thermodynamic equilibrium should be independent of which boundary $x_{\ell }\left(t\right)$ or $x_{s}\left(t\right)$ is chosen as the dynamical variable. Numerical and semi-analytical solutions will be used to illustrate that ΔL and $f_{s}$ or $f_{\ell }$ are independent of which boundary is set as the moving variable.

### 2.3. Absorbed/Released Thermal Energy

Numerical and semi-analytical solutions will be found for the dynamics of the phase transition during the charging or discharging process of a HTPCM. Mix boundary conditions can be used to produce full melting or solidification of the system. The solid phase will be completely melted through the following boundary conditions:(12)$$\begin{array}{c} T_{\ell }\left(x,t\right){|}_{x=x_{\ell }\left(t\right)}=T_{H},\\ T_{\ell }\left(x,t\right){|}_{x=\xi (t)}=T_{s}\left(x,t\right){|}_{x=\xi (t)}=T_{m},\;\;\;\;\mathrm{and}\\ k_{s}\;\frac{\partial \;T_{s}\left(x,t\right)}{\partial \;x}{|}_{x=x_{s}\left(t\right)}=0. \end{array}$$

Here the liquid domain is subjected to isothermal boundary conditions, where the left boundary is kept at the highest operating temperature $T_{H}$, and the solid domain is thermally isolated. Total thermal balance through the entire system introduces a correction through the apparent latent heat [29], which depends on the type of boundary conditions and the relative density difference between the liquid and solid phase, as follows:
(13a)$$L_{f}^{*}=L_{f}-\left(\frac{\rho_{s}}{\rho_{\ell }}-1\right)\;C_{s}\;(T_{m}-T_{s}\left(x,t\right){|}_{x=x_{s}\left(t\right)}),\;\;\mathrm{or}$$
(13b)$$L_{f}^{*}=L_{f}+\left(1-\frac{\rho_{\ell }}{\rho_{s}}\right)\;C_{\ell }\;\left(T_{H}-T_{m}\right).$$

On the one hand, Equation (13a) describes the apparent latent heat for a system where $x_{s}\left(t\right)$ is the moving boundary. On the other hand, Equation (13b) describes the apparent latent heat when $x_{\ell }\left(t\right)$ is the dynamical variable of motion [29]. The discharging process of the PCM is emulated through the following boundary conditions:(14)$$\begin{array}{c} k_{\ell }\;\frac{\partial \;T_{\ell }\left(x,t\right)}{\partial \;x}{|}_{x=x_{\ell }\left(t\right)}=0,\\ T_{\ell }\left(x,t\right){|}_{x=\xi (t)}=T_{s}\left(x,t\right){|}_{x=\xi (t)}=T_{m},\;\;\;\;\mathrm{and}\\ T_{s}\left(x,t\right){|}_{x=x_{s}\left(t\right)}=T_{C}. \end{array}$$

Here $T_{C}$ represents the lowest operating temperature during the process. The right boundary is in contact with a cold reservoir, which keeps the temperature constant at $x_{s}\left(t\right)$. According to the proposed model in Ref. [29], the apparent latent heat for a solidification process is given by
(15a)$$L_{f}^{*}=L_{f}-\left(\frac{\rho_{s}}{\rho_{\ell }}-1\right)\;C_{s}\;\left(T_{m}-T_{C}\right),\;\;\mathrm{or}$$
(15b)$$L_{f}^{*}=L_{f}+\left(1-\frac{\rho_{\ell }}{\rho_{s}}\right)\;C_{\ell }\;\left(T_{\ell }\left(x,t\right){|}_{x=x_{\ell }\left(t\right)}-T_{m}\right).$$

Equation (15a) predicts the apparent latent heat value when $x_{s}\left(t\right)$ is the moving boundary, and Equation (15b) is the apparent latent heat for a system where $x_{\ell }\left(t\right)$ is chosen as the dynamical variable. Melting and solidification rates are modified by the contributions from total thermal balance according to Equations (13a) and (13b) and Equations (15a) and (15b) [29].

The time evolution of the amount of thermal energy absorbed or released during the charging and discharging processes can be obtained through the enthalpy difference between the initial state of the system and the transient state at some time *t*. The thermal energy absorbed or released is obtained as follows:
(16a)$$\begin{array}{ccc} h(t)-h(0) & = & C_{\ell }\;\rho_{\ell }\;\left(\int_{x_{\ell }\left(t\right)}^{\xi (t)}T_{\ell }\left(x,t\right)\;dx-\int_{x_{\ell }\left(0\right)}^{\xi (0)}T_{\ell }\left(x,0\right)\;dx\right)+\\ & & C_{s}\;\rho_{s}\;\left(\int_{\xi (t)}^{x_{s}\left(t\right)}T_{s}\left(x,t\right)\;dx-\int_{\xi (0)}^{x_{s}\left(0\right)}T_{s}\left(x,0\right)\;dx\right),\;\;\;\;\mathrm{melting} \end{array}$$
(16b)$$\begin{array}{ccc} h(0)-h(t) & = & C_{\ell }\;\rho_{\ell }\;\left(\int_{x_{\ell }\left(0\right)}^{\xi (0)}T_{\ell }\left(x,0\right)\;dx-\int_{x_{\ell }\left(t\right)}^{\xi (t)}T_{\ell }\left(x,t\right)\;dx\right)+\\ & & C_{s}\;\rho_{s}\;\left(\int_{\xi (0)}^{x_{s}\left(0\right)}T_{s}\left(x,0\right)\;dx-\int_{\xi (t)}^{x_{s}\left(t\right)}T_{s}\left(x,t\right)\;dx\right),\;\;\;\;\mathrm{solidification} \end{array}$$

The apparent latent heat given by Equations (13a) and (13b) and Equations (15a) and (15b) takes into account the extra energy transfer during the phase transition, which is hidden by applying a local thermal balance at the interface. The corrections introduced through the apparent latent heat may change the predicted amount of melted (solidified) solid (liquid) during the phase change process when compared to the estimated predictions obtained by using the bulk latent heat in Equations (5a) and (5b). However, the latent heat absorbed or released must be determined through the bulk latent heat of fusion $L_{f}$, as will be shown in the following section.

### 2.4. Finite Element Method

The problem depends on space *x* and time *t*; therefore, both directions are approximated. In this section, we present a hybrid method to solve the problem, where the FEM is used to solve the spatial part, and an implicit finite difference method is used to approximate the time derivatives. The space and time dependence of the temperature distributions in the liquid and solid phases will be explicitly used in this part of the section to avoid confusion during the description of the FEM used in this work.

#### 2.4.1. Space Discretization: Finite Element Method

The region $\Omega =\{\Omega_{\ell }\cup \Omega_{s}\}$ represents the domain where the problem must be solved, such that $\Omega_{\ell }$ and $\Omega_{s}$ corresponds to the domain of the liquid and solid phase, respectively. For example, the regions $\Omega_{\ell }$ ($\Omega_{s}$) will be discretized into *n* (*m*) cubic elements, in such a way that
(17)$$\begin{array}{c} \Omega_{\ell }=\left[x_{1}^{\ell },x_{2}^{\ell },\cdot \cdot \cdot ,x_{3n+1}^{\ell }\right],\;\mathrm{with}\;x_{1}^{\ell }=0\;\mathrm{and}\;x_{3n+1}^{\ell }=\xi \left(t\right)\\ \Omega_{s}=\left[x_{1}^{s},x_{2}^{s},\cdot \cdot \cdot ,x_{3m+1}^{s}\right],\;\mathrm{with}\;x_{1}^{s}=\xi \left(t\right)\;\mathrm{and}\;x_{3m+1}^{s}=L\left(t\right). \end{array}$$

Here $x_{q}^{\ell }\;(q=1,2,\cdot \cdot \cdot ,3n+1$) and $x_{p}^{s}\;(p=1,2,\cdot \cdot \cdot ,3m+1$) are the coordinates of the nodes at the liquid and solid phases, respectively. The FEM formulation is the same for both phases; therefore, the subscript *i* that appears in Equation (4) will be omitted.

Suppose that we look for the approximate solution $\tilde{T}\left(x,t\right)$ at some time *t* within certain space of functions H with dimension 4; and $\Psi_{1},\Psi_{2},\cdot \cdot \cdot ,\Psi_{4}$ constitutes a basis of H. Then, the temperature field $\tilde{T}\left(x,t\right)$ can be written as follows:(18)$$\tilde{T}\left(x,t\right)=\sum_{d=1}^{4}{\hat{T}}_{d}^{\left(e\right)}\left(t\right)\Psi_{d}\left(x\right)\;,$$
where ${\hat{T}}_{d}^{\left(e\right)}\left(t\right)$ is the temperature at node *d*, associated with element *e*.

Applying the weighted residual method and Galerkin formulation [30,31], Equation (4) becomes
(19)$$\int_{\Omega }\left(\alpha \frac{\partial^{2}\tilde{T}\left(x,t\right)}{\partial x^{2}}-\frac{\partial \tilde{T}\left(x,t\right)}{\partial t}\right)v\left(x\right)\;dx=0\;,$$
where v(x) represents the test function. Integrating Equation (19) by parts, the following is obtained
(20)$$\int_{\Omega }-\alpha \frac{\partial v(x)}{\partial x}\frac{\partial \tilde{T}\left(x,t\right)}{\partial x}dx\;+\;{\left[\alpha \;v\left(x\right)\frac{\partial \tilde{T}\left(x,t\right)}{\partial x}\right]}_{\Omega }\;-\;\int_{\Omega }v\left(x\right)\frac{\partial \tilde{T}\left(x,t\right)}{\partial t}dx=0\;.$$

The integrals over Ω can be expressed as the sum of integrals over each element, and Equation (20) becomes
(21)$$\sum_{e=1}^{k}\int_{x_{e}}^{x_{e+1}}-\alpha \frac{\partial v(x)}{\partial x}\frac{\partial \tilde{T}\left(x,t\right)}{\partial x}dx\;+\;\sum_{e=1}^{k}{\left[\alpha v\left(x\right)\frac{\partial \tilde{T}\left(x,t\right)}{\partial x}\right]}_{x_{e}}^{x_{e+1}}\;-\;\sum_{e=1}^{k}\int_{x_{e}}^{x_{e+1}}v\left(x\right)\frac{\partial \tilde{T}\left(x,t\right)}{\partial t}dx=0\;,$$
where k=n or k=m, depending on the phase. Taking $v\left(x\right)=\Psi^{\;T}\left(x\right)$ where the T represents the transpose, and substituting Equation (18) into Equation (21) the following is obtained:(22)$$\sum_{e=1}^{k}K^{\left(e\right)}{\hat{T}}^{\left(e\right)}\left(t\right)+\sum_{e=1}^{k}M^{\left(e\right)}{\left({\hat{T}}^{\left(e\right)}\left(t\right)\right)}^{{}^{\prime }}=\sum_{e=1}^{k}F^{\left(e\right)}\;,$$
where
(23)$$\begin{array}{ccc} K^{\left(e\right)} & = & \alpha \int_{x_{e}}^{x_{e+1}}\frac{\partial \Psi^{\;T}\left(x\right)}{\partial x}\frac{\partial \Psi (x)}{\partial x}dx\;, \end{array}$$
(24)$$\begin{array}{ccc} M^{\left(e\right)} & = & \int_{x_{e}}^{x_{e+1}}\Psi^{\;T}\left(x\right)\Psi \left(x\right)\;dx\;, \end{array}$$
(25)$$\begin{array}{ccc} F^{\left(e\right)} & = & \alpha \Psi^{\;T}\left(x\right){\left[\frac{\partial \tilde{T}\left(x,t\right)}{\partial x}\right]}_{x_{e}}^{x_{e+1}}\;. \end{array}$$

Here ${\left({\hat{T}}^{\left(e\right)}\left(t\right)\right)}^{{}^{\prime }}$ is the time derivative of the temperature at each node. $K^{\left(e\right)},M^{\left(e\right)}$, and $F^{\left(e\right)}$ represent the element stiffness matrix, element mass matrix, and the element flux vector, respectively.

After assembling the local matrices the following global representation of the problem is obtained:(26)$$K\hat{T}\left(t\right)+M{\left(\hat{T}\left(t\right)\right)}^{{}^{\prime }}=F\;,$$
where K is the global stiffness matrix and M is the global mass matrix. On the other hand, F is the global load vector and $\hat{T}\left(t\right)={\left[{\hat{T}}_{1}\left(t\right),{\hat{T}}_{2}\left(t\right),\cdot \cdot \cdot ,{\hat{T}}_{q}\left(t\right)\right]}^{T}$ is the global temperature vector and represents the temperature at each node $\left[x_{1}^{i},x_{2}^{i},\cdot \cdot \cdot ,x_{q}^{i}\right]$ at time *t*, with i=ℓ(s) for the liquid (solid) phase. Dimension *q* is related to the elements number k=n(m) used to discretize $\Omega_{i}$ and matches the total number of nodes at each phase.

#### 2.4.2. Shape Functions

Once the region $\Omega_{i}$ is discretized, the shape functions Ψ(x), which will be used to approximate the solution $\tilde{T}\left(x,t\right)$, are selected. These functions are piecewise polynomials defined at each element. Lagrange functions are the most common type of shape functions in finite element analysis. In this work, we will use cubic Lagrange functions as shape functions for the FEM implementation. The cubic approach needs four points at every element *e*: one at each end of the interval and two more points equally spaced. So, for each element *e*, there are four equidistant nodes when using this approach. Four basic functions ($N_{1}\left(x\right)$, $N_{2}\left(x\right)$, $N_{3}\left(x\right)$, and $N_{4}\left(x\right)$) are needed on each element *e*. So, $K^{\left(e\right)}$ and $M^{\left(e\right)}$ are 4×4 square matrices and $F^{e}={\left[F_{1},\;F_{2},\;F_{3},\;F_{4}\right]}^{T}$. Dimension q=3n+1(3m+1). A higher degree of the shape function implies more exact results but also bigger dimensions for the system Equation (26). Cubic Lagrange shape functions for element *e* are
(27)$$\begin{array}{ccc} N_{1}\left(x\right) & = & \frac{\left(x-x_{2}\right)\left(x-x_{3}\right)\left(x-x_{4}\right)}{\left(x_{1}-x_{2}\right)\left(x_{1}-x_{3}\right)\left(x_{1}-x_{4}\right)}, \end{array}$$
(28)$$\begin{array}{ccc} N_{2}\left(x\right) & = & \frac{\left(x-x_{1}\right)\left(x-x_{3}\right)\left(x-x_{4}\right)}{\left(x_{2}-x_{1}\right)\left(x_{2}-x_{3}\right)\left(x_{2}-x_{4}\right)}, \end{array}$$
(29)$$\begin{array}{ccc} N_{3}\left(x\right) & = & \frac{\left(x-x_{1}\right)\left(x-x_{2}\right)\left(x-x_{4}\right)}{\left(x_{3}-x_{1}\right)\left(x_{3}-x_{2}\right)\left(x_{3}-x_{4}\right)}, \end{array}$$
(30)$$\begin{array}{ccc} N_{4}\left(x\right) & = & \frac{\left(x-x_{1}\right)\left(x-x_{2}\right)\left(x-x_{3}\right)}{\left(x_{4}-x_{1}\right)\left(x_{4}-x_{2}\right)\left(x_{4}-x_{3}\right)}. \end{array}$$

Let $N=[N_{1}\left(x\right),N_{2}\left(x\right),N_{3}\left(x\right),N_{4}\left(x\right)]$. Substituting Ψ(x)=N in Equations (23)–(25) we can compute local matrices $K^{\left(e\right)}$, $M^{\left(e\right)}$ and local vector $F^{\left(e\right)}$:$$\begin{array}{ccc} K^{\left(e\right)} & = & \alpha \int_{x_{e}}^{x_{e+1}}\frac{\partial N^{\;T}\left(x\right)}{\partial x}\frac{\partial N(x)}{\partial x}dx\;,\\ M^{\left(e\right)} & = & \int_{x_{e}}^{x_{e+1}}N^{\;T}\left(x\right)N\left(x\right)\;dx\;,\\ F^{\left(e\right)} & = & \alpha N^{\;T}\left(x\right){\left[\frac{\partial \tilde{T}\left(x,t\right)}{\partial x}\right]}_{x_{e}}^{x_{e+1}}\;. \end{array}$$

The integrals over each element are calculated by using Gaussian quadrature.

#### 2.4.3. Time Discretization: Implicit Finite Difference Scheme

Through the FEM, the differential problem Equation (4) has been transformed into a system of ordinary differential Equations (Equation 26). These equations must be solved for each time value *t*. The time derivative of the temperature ${\left(\hat{T}\left(t\right)\right)}^{{}^{\prime }}$ that appears in Equation (Equation 26) will be approximated by using an implicit finite difference scheme. The temperature ${\hat{T}}^{j}$ represents the temperature distribution or global temperature vector at the jth time level. The time derivative of the temperature ${\left(\hat{T}\left(t\right)\right)}^{{}^{\prime }}$ is approximated by using a backward difference in time as follows:(31)$${\left(\hat{T}\left(t\right)\right)}^{\prime }=\frac{\partial \hat{T}\left(t\right)}{\partial t}\approx \frac{{\hat{T}}^{j}-{\hat{T}}^{j-1}}{\Delta t}\;,$$
where Δt is the time step and *j* represents the *j*th time level. Substitution of Equation (31) in Equation (26) leads to the following system of equations
(32)$$\left(M+K\;\Delta t\right){\hat{T}}^{j}=M{\hat{T}}^{j-1}+F\;\Delta t\;.$$

The system of equations must be solved at each time level *j*. The temperature distribution ${\hat{T}}^{j-1}$ corresponds to the temperature values at the *j*th−1 time level, which are determined in a previous step. The temperatures at each node are obtained by using the temperature values from the previous step, and the system of equations is solved in each phase.

### 2.5. Refined Heat Balance Integral Method

The equations of motion for the interface position, total mass conservation, and the heat equation will be solved through a cubic Lagrange implementation of the FEM and a refined heat balance integral method (RHBIM). The energy absorbed or released by the PCM will be determined through the numerical (FEM) and semi-analytical (RHBIM) solutions to the model based on a local energy balance at the interface and the proposed model that considers total thermal balance.

The RHBIM consists of proposing polynomial temperature profiles in each phase with time dependent coefficients that are found by solving the resulting set of ordinary differential equations in time. The method is described in Refs. [13,14], and a brief description of the semi-analytical method will be given in this part of the section. The liquid and solid domains are divided into *m* and *n* regions, respectively. Quadratic temperature profiles in the spatial variable *x* are proposed at each region. Continuity and smoothness of the temperature distributions between adjacent regions are imposed. The temperature profiles within each phase and region are given by
(33)$$\begin{array}{ccc} T_{\ell_{i}}\left(x,t\right) & = & a_{i}\left(t\right)\;\left(x-x_{j}\right)+b_{i}\left(t\right)\;{\left(x-x_{j}\right)}^{2}+T_{i}\left(t\right),\;\;\;\;\mathrm{for}\;\;\;\;x_{\ell }\left(t\right)\leq \;x\leq \xi \left(t\right)\\ T_{s_{j}}\left(x,t\right) & = & c_{j}\left(t\right)\;\left(x-x_{j}\right)+d_{j}\left(t\right)\;{\left(x-x_{j}\right)}^{2}+T_{j}\left(t\right),\;\;\;\;\mathrm{for}\;\;\;\;\xi \left(t\right)\leq \;x\leq \;x_{s}\left(t\right). \end{array}$$

Time-dependent coefficients $a_{i}\left(t\right)$, $b_{i}\left(t\right)$, and $T_{i}\left(t\right)$ at each region *i* within the liquid domain and $c_{j}\left(t\right)$, $d_{j}\left(t\right)$, and $T_{j}\left(t\right)$ at each region *j* within the solid domain are used to determine the time evolution of the temperature profiles. Continuity and smoothness, along with the isothermal boundary condition at the interface and the boundary conditions given by Equations (12) and (14), are used to find the time-dependent coefficients $a_{i}\left(t\right)$, $c_{j}\left(t\right)$, $T_{i}\left(t\right)$, and $T_{j}\left(t\right)$ in terms of $b_{i}\left(t\right)$ and $d_{j}\left(t\right)$. The interface position is found by substitution of the temperature profiles given through Equation (33) into the equation of motion for the interface, described through Equation (5a) or Equation (5b). Total thermal balance is considered by using the corresponding expression for the apparent latent heat in Equation (5a) or Equation (5b). Additionally, local thermal balance is considered by using the bulk latent heat in Equation (5a) or Equation (5b). The position of the dynamical variable $x_{s}\left(t\right)$ or $x_{\ell }\left(t\right)$ is found through total mass conservation given by Equations (6a) and (6b). Finally, the heat equation is averaged over the liquid and solid domains by using the temperature profiles given by Equation (33), resulting in a set of m+n ordinary differential equations (ODEs) in time for the coefficients $b_{i}\left(t\right)$ and $d_{j}\left(t\right)$. A first-order approximation for the time derivatives that appear in the resulting system of ODEs is used to obtain a linear system of algebraic equations for ξ(t), $x_{s}\left(t\right)$$\left(x_{\ell }\left(t\right)\right)$, $b_{i}\left(t\right)$, and $d_{j}\left(t\right)$ at each region in the next time level t+Δt. The linear system of equations was solved by using time steps of Δt=0.5μs for the melting of ${\mathrm{KNO}}_{3}$ salt and Δt=0.1μs for the rest of the numerical examples. The dynamics of the phase transition were determined until the mass fraction of melted (solidified) solid (liquid) $f_{s}\geq \;0.999$ ($f_{\ell }\geq \;0.999$).

## 3. Results and Discussion

${\mathrm{KNO}}_{3}$ and the euctectic ${\mathrm{KNO}}_{3}/{\mathrm{NaNO}}_{3}$ salts are used as HTPCMs to find the thermal energy absorbed (released) during a charging (discharging) process. The thermodynamic properties of the salts used are summarized in Table 1 and are the same properties used in Ref. [29]. The specific heat capacities correspond to their values close to the melting temperature of the PCM, and the latent heat of fusion is obtained as $L_{f}=\left(C_{\ell }-C_{s}\right)\;T_{m}$.

### 3.1. Invariance of Solutions in Adiabatic Systems

The first part of this section is devoted to the discussion of the phase change process in adiabatic systems. In this section, the HTPCM considered is the ${\mathrm{KNO}}_{3}$ salt. A wide temperature range is used to highlight the breaking of the invariance when $x_{s}\left(t\right)$ or $x_{\ell }\left(t\right)$ is chosen as the moving boundary, according to the solutions obtained through a local energy balance at the interface. Two examples are shown where the initial energy of the liquid–solid sample produces melting on the one hand and solidification on the other hand. The initial temperature at $x_{\ell }\left(t\right)$$\left(x_{s}\left(t\right)\right)$ is $T_{H_{0}}=923\;K$ ($T_{C_{0}}=535\;K$) for the melting of ${\mathrm{KNO}}_{3}$ with an initial interface position ξ(0)=0.30m and initial size of L(0)=1.0m. The initial temperature at $x_{\ell }\left(t\right)$$\left(x_{s}\left(t\right)\right)$ is $T_{H_{0}}=680\;K$ ($T_{C_{0}}=300\;K$) for the solidification example, where ξ(0)=0.70m. These conditions are used to determine the initial temperature distribution in the liquid and solid domains as quadratic functions of the spatial variable *x*. The system is subjected to the adiabatic boundary conditions given by Equation (1) and the isothermal boundary condition at the liquid–solid interface, given by Equation (2). The total energy is a constant of the motion since the system is thermally isolated. The bar is expected to reach thermodynamic equilibrium where the system growth (shrinkage) and the fraction of melted (solidified) solid (liquid) is given by Equations (10) and (11), respectively.

Figure 1 shows the time evolution of ΔL and $f_{s}$ ($f_{\ell }$) for ${\mathrm{KNO}}_{3}$ according to the solutions obtained by assuming the classical local thermal balance at the interface. Figure 1a,b shows the system growth and fraction of melted solid $f_{s}$ upon melting of ${\mathrm{KNO}}_{3}$. The numerical solutions to the classical model when $x_{s}\left(t\right)$ or $x_{\ell }\left(t\right)$ is chosen as the moving boundary do not have the same time-dependent behavior for ΔL and $f_{s}$. These solutions are not invariant when the volume of the system is allowed to change from the right or left boundary. Additionally, it is realized that solutions for ΔL and $f_{s}$ ($f_{\ell }$) reach entirely different thermodynamic equilibrium values when the sample is fixed at the left boundary and when the sample is fixed at the right boundary. The lack of invariance of the numerical solutions to the classical model is in contradiction with the predicted thermodynamic equilibrium values shown through Equations (10) and (11). Figure 1c,d also illustrates this anomalous behavior when the ${\mathrm{KNO}}_{3}$ sample shrinks upon solidification of liquid in adiabatic systems.

Equation (5a) or Equation (5b) are solved through the FEM by imposing energy conservation through the apparent latent heat $L_{f}^{*}$ given by Equation (8a) or Equation (8b), whether $x_{s}\left(t\right)$ or $x_{\ell }\left(t\right)$ is the time-dependent boundary. Additionally, total mass conservation is imposed through Equation (6a) or Equation (6b). The numerical solutions to ΔL and the fraction of melted (solidified) solid (liquid) are independent of which boundary is chosen as the dynamical variable, as illustrated in Figure 2. According to the proposed model, the FEM solutions are observed to reach the predicted thermodynamic equilibrium values given by Equations (10) and (11). The solutions during the melting and solidification processes in ${\mathrm{KNO}}_{3}$ are independent of which boundary is chosen as the dynamical variable, as illustrated in Figure 2. Figure 2a,b show the FEM solutions when the initial conditions produce melting of ${\mathrm{KNO}}_{3}$. The time evolution of ΔL and the fraction of melted solid $f_{s}$ are independent of which boundary is chosen as the dynamical variable. Additionally the thermodynamic equilibrium values for $\Delta \;L_{\mathrm{eq}}=24.775735\;\mathrm{mm}$ and $f_{s_{\mathrm{eq}}}=0.910129$ are well reproduced by the FEM solution at t=115.74days as: $\Delta \;L_{\mathrm{eq}}^{\left(\mathrm{FEM}\right)}=24.593533\;\mathrm{mm}$ and $f_{s_{\mathrm{eq}}}^{\left(\mathrm{FEM}\right)}=0.903436\;\mathrm{mm}$. Figure 2c,d corresponds to the solidification example of ${\mathrm{KNO}}_{3}$ illustrated in Figure 1. The numerical solutions are also invariant and well behaved near the thermodynamic equilibrium state of the system. Finite element method solutions reach the predicted values at thermodynamic equilibrium $\Delta \;L_{\mathrm{eq}}=19.830973\;\mathrm{mm}$ and $f_{\ell_{\mathrm{eq}}}=0.756815$ at t=115.74days as: $\Delta \;L_{\mathrm{eq}}^{\left(\mathrm{FEM}\right)}=19.838997\;\mathrm{mm}$ and $f_{\ell_{\mathrm{eq}}}^{\left(\mathrm{FEM}\right)}=0.757121$.

Total energy in adiabatic systems is a constant of the motion. The initial energy of the system E(0) must be conserved throughout the melting or solidification process. Therefore, energy conservation can be used to determine the performance of the numerical solutions. An average energy error (AEE) for the total energy has been defined as follows for this purpose:(34)$$\mathrm{AEE}=\frac{\sum_{i=1}^{n}\left|E\left(0\right)-E\left(t_{i}\right)\right|}{n},$$
where E(0) is the initial energy of the system, $E(t_{i})$ is the total energy of the system at some time level $t_{i}$, and *n* is the number of time partitions.

Three versions of the FEM were implemented: the first one labeled as ${\mathrm{FEM}}^{1}$ uses linear Lagrange functions, ${\mathrm{FEM}}^{2}$ is an implementation of the FEM with quadratic Lagrange functions, and ${\mathrm{FEM}}^{3}$ uses the cubic Lagrange shape functions described in Section 2.4. Table 2 shows the AEE for each implementation of the FEM during melting and solidification of ${\mathrm{KNO}}_{3}$. Results are shown for ten elements that were used to discretize the entire spatial domain. The number of elements within the liquid and solid domains was varied during the phase change process. The number of elements at each phase and in a given time level was determined from the volume of the liquid and solid phases in order to have a total number of ten elements in the whole system with the same length. A total of $n=1\times \;10^{5}$ time partitions were used on each of the examples illustrated in Figure 1 and Figure 2 and Table 2. Increasing the size of *n* produces negligible changes to the AEE given by Equation (34) and shown in Table 2. Local matrices obtained for the ${\mathrm{FEM}}^{1}$, ${\mathrm{FEM}}^{2}$, and ${\mathrm{FEM}}^{3}$ implementations have dimensions of 2, 3, and 4, respectively. Increasing the degree of the shape functions produces lower values for the AEE, as expected. Table 2 illustrates how total energy conservation is best behaved through a ${\mathrm{FEM}}^{3}$ implementation of the FEM.

### 3.2. Energy Absorbed/Released

The liquid–solid system is subjected to mixed boundary conditions to produce melting on the one hand and solidification of the liquid phase on the other hand. Melting examples are designed through the boundary conditions given by Equation (12), and solidification of liquid phase is achieved by imposing the boundary conditions through Equation (14). The system will absorb (release) thermal energy until the solid (liquid) is almost completely melted (solidified). The total amount of energy absorbed or released by the PCM can be obtained through Equation (16a) or Equation (16b).

The sensible heat absorbed by a PCM can be conceived in four stages, as discussed in Refs. [14,25]. These stages can be used to show that latent heat is absorbed through the bulk latent heat $L_{f}$ of the PCM and not the apparent latent heat previously defined [29]. The first stage considers the thermal energy absorbed between t=0 and some time *t* by the initial mass of liquid as
(35)$$\Delta \;h_{1}=C_{\ell }\;\rho_{\ell }\int_{0}^{\xi (0)}T_{\ell }\left(x,t\right)\;dx-C_{\ell }\;\rho_{\ell }\int_{0}^{\xi (0)}T_{\ell }\left(x,0\right)\;dx,$$
where it is assumed that $x_{\ell }\left(t\right)$ is constant and equal to $x_{\ell }\left(t\right)=0$, and $x_{s}\left(t\right)$ is the dynamical variable used to impose total mass conservation. During a second stage, the amount of solid mass $\Delta \;m_{s}$ that will be eventually melted will increase its temperature from its initial value $T_{s}\left(x,0\right)$ to the melting temperature $T_{m}$, absorbing thermal energy as follows:(36)$$\Delta \;h_{2}=\Delta \;m_{s}\;C_{s}\;T_{m}-C_{s}\;\rho_{s}\int_{\xi (0)}^{x_{p}}T_{s}\left(x,0\right)\;dx,$$
where $x_{p}-\xi \left(0\right)$ is the volume of this amount of solid that will be melted between t=0 and any later time *t* and is related with $\Delta \;m_{s}$ as $\Delta \;m_{s}=\rho_{s}\;\left(x_{p}-\xi \left(0\right)\right)$. The value of $x_{p}$ can be obtained through mass conservation as follows:(37)$$\rho_{s}\;\left(x_{p}-\xi \left(0\right)\right)=\rho_{s}\;\left(x_{s}\left(0\right)-\xi \left(0\right)\right)-\rho_{s}\;\left(x_{s}\left(t\right)-\xi \left(t\right)\right);$$
then, solving for $x_{p}$ the following expression is obtained $x_{p}=\xi \left(t\right)-\Delta \;L$, where $\Delta \;L=x_{s}\left(t\right)-x_{s}\left(0\right)$. The third stage considers the energy absorbed by $\Delta \;m_{s}$ once it has transformed into liquid phase. Through this stage, this mass of liquid absorbs thermal energy from the melting temperature $T_{m}$, to the temperature of the liquid phase $T_{\ell }\left(x,t\right)$ at time *t*, as follows:(38)$$\Delta \;h_{3}=C_{\ell }\;\rho_{\ell }\int_{\xi (0)}^{\xi (t)}T_{\ell }\left(x,t\right)\;dx-\Delta \;m_{s}\;C_{\ell }\;T_{m},$$
where ξ(t)−ξ(0) is the volume of $\Delta \;m_{s}$, but now in its liquid state. Finally, the mass of solid that was not melted between the initial state and the state of the system at some later time *t*, will absorb thermal energy as sensible heat by raising its temperature from its initial value $T_{s}\left(x,0\right)$ to the temperature $T_{s}\left(x,t\right)$ at some later time *t* as follows:(39)$$\Delta \;h_{4}=C_{s}\;\rho_{s}\int_{\xi (t)}^{x_{s}\left(t\right)}T_{s}\left(x,t\right)\;dx-C_{s}\;\rho_{s}\int_{x_{p}}^{x_{s}\left(0\right)}T_{s}\left(x,0\right)\;dx.$$

Adding the contributions from each stage to the total sensible heat absorbed, the following expression is obtained:(40)$$\begin{array}{ccc} Q_{s} & = & C_{\ell }\;\rho_{\ell }\;\left(\int_{0}^{\xi (t)}T_{\ell }\left(x,t\right)\;dx-\int_{0}^{\xi (0)}T_{\ell }\left(x,0\right)\;dx\right)+\\ & & C_{s}\;\rho_{s}\;\left(\int_{\xi (t)}^{x_{s}\left(t\right)}T_{s}\left(x,t\right)\;dx-\int_{\xi (0)}^{x_{s}\left(0\right)}T_{s}\left(x,0\right)\;dx\right)-\left(\Delta \;m_{s}\;\left(C_{\ell }-C_{s}\right)\;T_{m}\right). \end{array}$$

The first two terms correspond to the total enthalpy absorbed by a melting PCM as shown by Equation (16a) for $x_{\ell }\left(t\right)$ constant and equal to zero, and $x_{s}\left(t\right)$ as the moving boundary. The last term is exactly the latent heat absorbed by the PCM during the phase change process, which is proportional to the bulk latent heat $L_{f}=\left(C_{\ell }-C_{s}\right)\;T_{m}$ and not the apparent latent heat. The above discussion shows that even though solidification or melting rates can be pictured through an apparent latent heat, the latent heat storage capacity of the PCM depends only on the bulk latent heat and not $L_{f}^{*}$. A similar analysis can be performed for solidification scenarios or when $x_{\ell }\left(t\right)$ is chosen as the dynamical variable.

The present work estimates the contributions from the sensible and latent heat stored or released through Equations (16a) and (16b), according to the numerical and semi-analytical solutions to the old and new models. Figure 3 shows the total energy absorbed and released by the ${\mathrm{KNO}}_{3}$ salt. Melting (solidification) examples are illustrated until the fraction of melted (solidified) solid (liquid) is $f_{s}\geq \;0.999$ ($f_{\ell }\geq \;0.999$). Numerical and semi-analytical solutions according to a local thermal balance at the interface and the proposed total thermal balance are shown. Melting of ${\mathrm{KNO}}_{3}$ is produced through the boundary conditions given by Equation (12). The initial position of the liquid–solid interface is ξ(0)=0.05m in Figure 3a,b. The temperature at $x_{\ell }\left(t\right)$ is kept at a constant value of $T_{H}=680\;K$, and the initial temperature at the right boundary is $T_{C_{0}}=240\;K$. Figure 3a,b shows the numerical and semi-analytical estimations of the total energy absorbed by the salt according to both models and for each case, when $x_{s}\left(t\right)$ or $x_{\ell }\left(t\right)$ is the moving boundary. The relative difference between the estimations of both models is practically negligible upon melting of the solid phase. Alternatively, significant differences are observed when the salt is releasing thermal energy. Figure 3c,d illustrates the numerical and semi-analytical solutions when the liquid experiences solidification and the system is subjected to the boundary conditions given by Equation (14). Initially, a small volume of solid is considered with ξ(0)=0.95m. The right boundary is kept at a constant temperature value of $T_{C}=240\;K$ and the initial temperature at $x_{\ell }\left(t\right)$ is $T_{H_{0}}=680\;K$.

The relative percent difference (RPD) between the old and new models in the total energy absorbed by the ${\mathrm{KNO}}_{3}$ salt is shown in Table 3. The small difference can be understood in terms of the apparent latent heat, which depends on the boundary conditions. On the one hand, according to Equation (13a) the corrections to the bulk latent heat are proportional to the difference between the melting temperature and the temperature at the right boundary. The temperature at $x_{s}\left(t\right)$ is expected to approach $T_{m}$ as the solid melts; therefore, this term becomes smaller as the system evolves in time. On the other hand, the contributions from the apparent latent heat when $x_{\ell }\left(t\right)$ is the moving boundary can be negligible according to Equation (13b) if the temperature at this boundary is close to $T_{m}$. The RPD between the predictions obtained from the old and new models was determined as follows:(41)$$\mathrm{RPD}=\frac{\left|\Delta \;h_{N}^{\left(x_{i}\right)}-\Delta \;h_{O}^{\left(x_{i}\right)}\right|}{(\Delta \;h_{N}^{\left(x_{i}\right)}+\Delta \;h_{O}^{\left(x_{i}\right)})/2}\times \;100\;\%,$$
where $x_{i}$ with i=ℓ,s corresponds to which boundary is considered as the dynamical variable.

Higher RPD values are expected for the solidification of ${\mathrm{KNO}}_{3}$ as illustrated in Table 4 since the temperature at the right boundary is much smaller than the melting point of the salt. According to Equation (15a), when the system releases energy and $x_{s}\left(t\right)$ is the moving boundary, the corrections introduced through the apparent latent heat are proportional to $T_{m}-T_{C}$. The example shown in Figure 3c illustrates the solidification of liquid at a very low temperature value of $T_{C}=240\;K$, well below the melting point of the salt. However, when $x_{\ell }\left(t\right)$ is the dynamical variable, the contributions from the apparent latent heat are much smaller since $L_{f}^{*}$ approaches the bulk latent heat of fusion $L_{f}$ as the system evolves in time [29].

Finally, the energy absorbed and released by the ${\mathrm{KNO}}_{3}/{\mathrm{NaNO}}_{3}$ salt is estimated through the numerical and semi-analytical solutions to both models discussed in this work. The RPD between both models is highly related to the thermodynamic properties of the material and the boundary conditions. The salt can be exposed to higher temperature values, and due to its lower melting temperature as shown in Table 1, the expected difference between both models should be higher when the salt is absorbing thermal energy. Figure 4 shows the numerical and semi-analytical solutions for the thermal energy absorbed and released by the ${\mathrm{KNO}}_{3}/{\mathrm{NaNO}}_{3}$ salt. The charging process is shown in Figure 4a,b. The initial interface position during the melting process of the salt is 0.05m. The temperature at $x_{\ell }\left(t\right)$ is kept constant at $T_{H}=866\;K$, well above the melting temperature $T_{m}=496\;K$ of the salt. The left boundary is thermally isolated, and its initial temperature is $T_{C_{0}}=300\;K$. The discharging process starts with a small volume of solid, where the initial interface position is ξ(0)=0.95m. Heat is removed from the right boundary which is kept at a constant temperature value of $T_{C}=300\;K$, and the left boundary is initially set at $T_{H_{0}}=300\;K$. The solutions for the thermal energy released during the discharging process are shown in Figure 4c,d.

According to Equation (13b), due to the high difference between the melting temperature of the salt and a maximum charging temperature $T_{H}=866\;K$, the contributions from the apparent latent heat are significant in the example shown in Figure 4b. The RPD for the energy absorbed by the salt and according to both models is shown in Table 5. The apparent latent heat, according to the new model, predicts lower energy absorption rates than those estimated by assuming local thermal balance at the interface. This behavior is observed in Figure 4b, which according to Equation (13b) the PCM should take longer time values to absorb thermal energy until the sample is completely melted. The asymptotic time behavior of the apparent latent heat when $x_{s}\left(t\right)$ is the moving boundary predicts lower RPD values, as illustrated in Figure 4a and Table 5.

Lower RPD values are expected upon solidification of the ${\mathrm{KNO}}_{3}/{\mathrm{NaNO}}_{3}$ salt since thermal energy is drained from the right boundary at temperatures close to the melting point of the salt. According to Equation (15a) the contributions from total thermal balance are significantly lower due to the operating temperature value at the right boundary. The effects of total thermal balance on the released energy are even smaller when $x_{\ell }\left(t\right)$ is the moving boundary since the apparent latent heat approaches asymptotically to the bulk latent heat $L_{f}$, as illustrated in Table 6.

Charging and discharging times are also an important parameter when HTPCMs are used as backup systems in thermoelectric generation applications. The proposed model estimates different energy densities of the salts used in this work and different charging or discharging times. According to the FEM and RHBIM solutions for the thermal energy absorbed or released by the ${\mathrm{KNO}}_{3}$ salt, the highest RPD between both models is found during a discharging process. The result is consistent with Equation (15a) that predicts significantly smaller energy releasing rates. Then according to the new model, the PCM should release thermal energy at a faster rate. The behavior can be observed in Figure 3c. The maximum RPD between discharging times according to the numerical and semi-analytical solutions is 9.38% and 9.17%, respectively. Thermodynamic properties and operating temperatures used for the example shown in Figure 4 increase the RPD between both models in a charging process. According to the proposed model, the apparent latent heat is increased by total thermal balance, as shown through Equation (13b). Maximum RPD between charging times is expected, as shown in Figure 4b. Equation (13b) predicts lower energy absorption rates, as illustrated in Figure 4b. The estimated RPD between charging times according to the numerical and semi-analytical solutions in this example is 9.76% and 9.58%, respectively.

## 4. Conclusions

Volume changes produced by the density difference between liquid and solid phases are taken into account through imposing total mass, as a constant of the motion. Conservation of total mass is incorporated through an equation of motion for the right or left boundary of the system, allowing the sample to expand or shrink during the phase transition. Thermodynamic equilibrium values for the volume changes and fraction of melted (solidified) solid (liquid) in adiabatic systems have been established and shown to be independent of the direction in which volume changes are taking place. The solutions obtained from the local thermal balance at the interface are not invariant in adiabatic systems and predict different thermodynamic equilibrium values, which is in contradiction with the expected behavior. Total energy conservation was imposed as a constant of the motion in adiabatic systems. Energy conservation introduced higher-order corrections to the equation of motion for the interface that were incorporated through an apparent latent heat. The numerical and semi-analytical solutions, considering energy transfer rates through the apparent latent heat, are invariant and consistent with previously established thermodynamic equilibrium values in adiabatic systems.

Thermal balance through the system was considered in samples with other types of boundary conditions. The corrections from total thermal balance were introduced by the apparent latent heat. These corrections are deeply related with the type of boundary conditions, the melting temperature of the material, and the relative density difference between liquid and solid phases. Significant differences between the local thermal balance at the interface and the total thermal balance in the energy absorbed (released) are found in HTPCMs. The magnitude of the correction introduced by total thermal balance to the thermal energy absorbed (released) is increased by the thermodynamic properties of HTPCMs and typical operating temperatures. The apparent latent heat only introduces a change to the energy absorption (release) rates, but it can not be conceived as a correction to the bulk latent heat of the PCM. Finally, charging (discharging) times are also modified by considering the proposed total thermal balance. Relative percent differences of 9.58% between both models in the amount of time needed to melt (solidify) the solid (liquid) phase are found. 

## Figures and Tables

**Figure 1 molecules-26-00365-f001:**
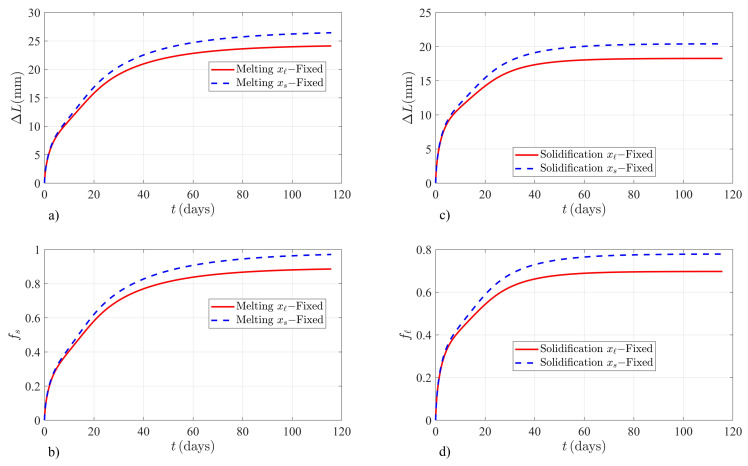
Finite element method solutions to the classical model for a liquid–solid sample of ${\mathrm{KNO}}_{3}$ salt subjected to the adiabatic boundary conditions given by Equation (1). (**a**) Time evolution of ΔL in mm and (**b**) fraction of melted solid $f_{s}$, for melting of ${\mathrm{KNO}}_{3}$ when ξ(0)=0.30m. (**c**) System shrinkage ΔL in mm and (**d**) $f_{\ell }$, for the solidification case of ${\mathrm{KNO}}_{3}$ when ξ(0)=0.70m.

**Figure 2 molecules-26-00365-f002:**
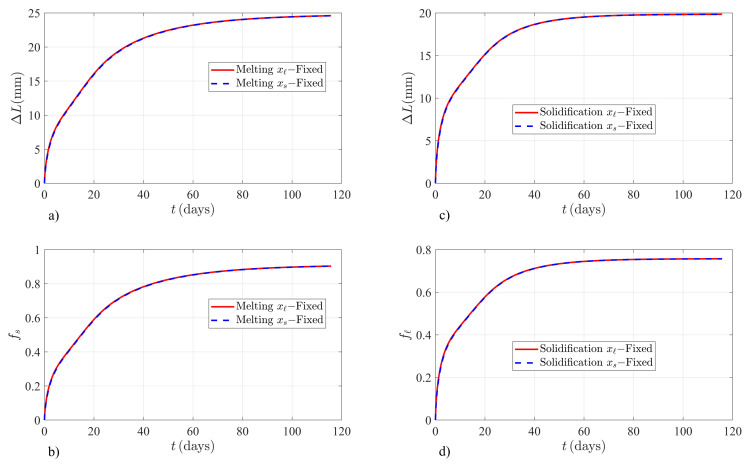
Finite element method solutions to the proposed model for the examples shown in Figure 1. (**a**) Time evolution of the system growth ΔL in mm and (**b**) fraction of melted solid $f_{s}$, when the initial interface position is ξ(0)=0.30m. (**c**) Shrinkage ΔL in mm and (**d**) fraction of solidified liquid $f_{\ell }$, when the initial interface position is ξ(0)=0.70m. Solutions are invariant and reach the correct thermodynamic equilibrium values shown through Equations (10) and (11).

**Figure 3 molecules-26-00365-f003:**
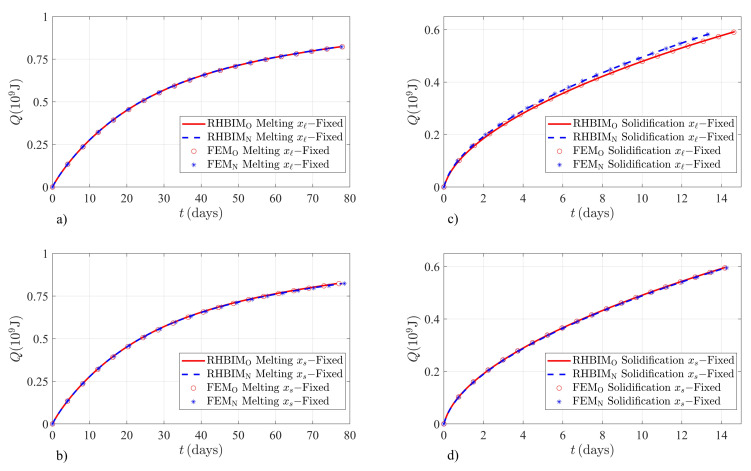
Finite element method and RHBIM solutions to the total energy absorbed and released by the ${\mathrm{KNO}}_{3}$ salt. Solutions obtained from the old and new models are shown in each case. (**a**) Time evolution of the energy absorbed Q=h(t)−h(0) in GJ when $x_{\ell }\left(t\right)$ is constant in time. Solutions corresponding to the old model were obtained from Equations (4), (5a), and (6a), and according to the new model by using the apparent latent heat given by Equation (13a). (**b**) Thermal energy absorbed *Q* when $x_{s}\left(t\right)$ is fixed. Solutions from local thermal balance are obtained through Equations (4), (5b), and (6b), and the contributions from total thermal balance were considered through the apparent latent heat given by Equation (13b). (**c**) Energy released Q=h(0)−h(t) during solidification of the liquid phase. Thermal energy released according to the new model is estimated from the solutions obtained through the apparent latent heat given by Equation (15a). (**d**) Thermal energy released when $x_{s}\left(t\right)$ is fixed in time. Predictions from the solutions according to the new model are obtained through the apparent latent heat given by Equation (15b).

**Figure 4 molecules-26-00365-f004:**
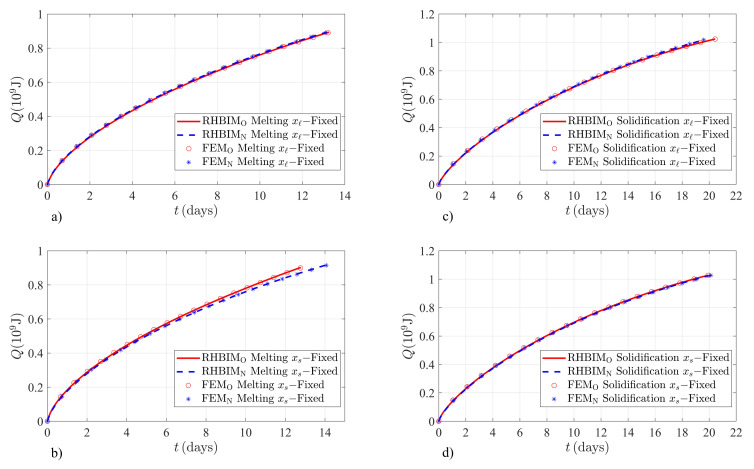
Finite element method and RHBIM solutions to the total energy absorbed and released by the ${\mathrm{KNO}}_{3}/{\mathrm{NaNO}}_{3}$ salt. Solutions obtained from the old and new models are shown in each case. (**a**) Time evolution of the energy absorbed Q=h(t)−h(0) in GJ when $x_{\ell }\left(t\right)$ is fixed in time. (**b**) Thermal energy absorbed *Q* when $x_{s}\left(t\right)$ is fixed. (**c**) Released energy Q=h(0)−h(t) during a discharge process when the left boundary is fixed in time. (**d**) Thermal energy released when $x_{s}\left(t\right)$ is constant in time.

**Table 1 molecules-26-00365-t001:** Thermodynamic properties of ${\mathrm{KNO}}_{3}$ and ${\mathrm{KNO}}_{3}/{\mathrm{NaNO}}_{3}$ salts. Specific heat capacity values for the liquid and solid phase are close to the melting temperature of the salt.

Salt	Ref.	$T_{m}$	$k_{\ell }$	$k_{s}$	$C_{\ell }$	$C_{s}$	$\rho_{\ell }$	$\rho_{s}$
		**K**	W/m·K		kJ/kg·K		$\mathrm{kg}/m^{3}$	
${\mathrm{KNO}}_{3}$	[9,10]	607	0.425	0.5	1.517	1.4	1800	1870
${\mathrm{KNO}}_{3}/{\mathrm{NaNO}}_{3}$	[10]	496	0.8	1.0	1.5	1.43	2096	2192

**Table 2 molecules-26-00365-t002:** Average energy error (AEE) in Giga-Joules (GJ), upon melting and solidification of ${\mathrm{KNO}}_{3}$ in a sample with adiabatic boundary conditions according to the linear, quadratic, and cubic implementations of the FEM. The initial energy of the system during the solidification (melting) example is E(0)=1.5692GJ (E(0)=1.6942GJ). The AEEs are approximately three, four, and five orders of magnitude smaller than E(0) according to the ${\mathrm{FEM}}^{1}$, ${\mathrm{FEM}}^{2}$, and ${\mathrm{FEM}}^{3}$ implementations, respectively. The AEE given by Equation (34) for each implementation of the FEM is illustrated when $x_{s}\left(t\right)$ or $x_{\ell }\left(t\right)$ is the moving boundary.

	Melting		Solidification	
	${\mathrm{AEE}}^{\left(x_{s}\right)}$	${\mathrm{AEE}}^{\left(x_{\ell }\right)}$	${\mathrm{AEE}}^{\left(x_{s}\right)}$	${\mathrm{AEE}}^{\left(x_{\ell }\right)}$
${\mathrm{FEM}}^{1}$	2.7754 × 10 ${}^{-3}$	4.0360 × 10 ${}^{-3}$	1.5241 × 10 ${}^{-4}$	1.1485 × 10 ${}^{-3}$
${\mathrm{FEM}}^{2}$	3.5981 × 10 ${}^{-4}$	3.7630 × 10 ${}^{-4}$	1.3639 × 10 ${}^{-5}$	2.7861 × 10 ${}^{-5}$
${\mathrm{FEM}}^{3}$	1.3529 × 10 ${}^{-5}$	4.7084 × 10 ${}^{-5}$	7.9583 × 10 ${}^{-6}$	6.2626 × 10 ${}^{-6}$

**Table 3 molecules-26-00365-t003:** Melting of ${\mathrm{KNO}}_{3}$ according to the numerical and semi-analytical solutions to the absorbed energy Δh=h(t)−h(0) estimated through the new and old models. The RPDs are obtained from the example shown in Figure 3a,b.

	FEM			RHBIM		
**Days**	$\Delta h_{p}^{\left(x_{s}\right)}\;\left(\mathrm{GJ}\right)$	$\Delta h_{c}^{\left(x_{s}\right)}\;\left(\mathrm{GJ}\right)$	RPD%	$\Delta h_{p}^{\left(x_{s}\right)}\;\left(\mathrm{GJ}\right)$	$\Delta h_{c}^{\left(x_{s}\right)}\;\left(\mathrm{GJ}\right)$	RPD%
15.42	0.3769	0.3763	0.1593	0.3794	0.3788	0.1582
30.84	0.5751	0.5836	1.4672	0.5777	0.5768	0.1560
46.26	0.6917	0.6906	0.1592	0.6938	0.6928	0.1442
61.68	0.7664	0.7656	0.1044	0.7684	0.7676	0.1042
77.10	0.8203	0.8197	0.0732	0.8221	0.8215	0.0730
	**FEM**			**RHBIM**		
**Days**	$\Delta h_{p}^{\left(x_{\ell }\right)}\;\left(\mathrm{GJ}\right)$	$\Delta h_{c}^{\left(x_{\ell }\right)}\;\left(\mathrm{GJ}\right)$	RPD%	$\Delta h_{p}^{\left(x_{\ell }\right)}\;\left(\mathrm{GJ}\right)$	$\Delta h_{c}^{\left(x_{\ell }\right)}\;\left(\mathrm{GJ}\right)$	RPD%
15.28	0.3755	0.3757	0.0532	0.3780	0.3782	0.0529
30.56	0.5731	0.5736	0.0872	0.5758	0.5859	1.7388
45.84	0.6893	0.6907	0.2029	0.6915	0.6930	0.2166
61.12	0.7635	0.7663	0.3660	0.7655	0.7683	0.3651
76.40	0.8168	0.8208	0.4885	0.8187	0.8227	0.4873

**Table 4 molecules-26-00365-t004:** Released energy Δh=h(0)−h(t) during solidification of ${\mathrm{KNO}}_{3}$ according to the numerical and semi-analytical solutions to the new and old models. The RPDs are obtained from the example shown in Figure 3c,d.

	FEM			RHBIM		
**Days**	ΔhN(xs)(GJ)	ΔhO(xs)(GJ)	RPD%	ΔhN(xs)(GJ)	ΔhO(xs)(GJ)	RPD%
3.90	0.3675	0.3651	0.6552	0.3624	0.3601	0.6366
7.80	0.5893	0.5860	0.5615	0.5858	0.5826	0.5478
11.70	0.7628	0.7580	0.6312	0.7606	0.7559	0.6198
15.60	0.9021	0.8948	0.8125	0.9075	0.8938	1.5211
19.50	1.0154	1.0019	1.3384	1.0155	1.0020	1.3382
	**FEM**			**RHBIM**		
**Days**	ΔhN(xℓ)(GJ)	ΔhO(xℓ)(GJ)	RPD%	ΔhN(xℓ)(GJ)	ΔhO(xℓ)(GJ)	RPD%
3.98	0.3750	0.3754	0.1066	0.3750	0.3746	0.1067
7.96	0.5993	0.6043	0.8308	0.5963	0.6011	0.8017
11.94	0.7734	0.7789	0.7086	0.7717	0.7771	0.6973
15.92	0.9119	0.9174	0.6013	0.9113	0.9168	0.6017
19.90	1.0221	1.0268	0.4587	1.0227	1.0275	0.4682

**Table 5 molecules-26-00365-t005:** RPD between both models during a charging process of ${\mathrm{KNO}}_{3}/{\mathrm{NaNO}}_{3}$ salt and according to the numerical and semi-analytical solutions to the absorbed energy Δh=h(t)−h(0).

	FEM			RHBIM		
**Days**	$\Delta h_{N}^{\left(x_{s}\right)}\;\left(\mathrm{GJ}\right)$	$\Delta h_{O}^{\left(x_{s}\right)}\;\left(\mathrm{GJ}\right)$	RPD%	$\Delta h_{N}^{\left(x_{s}\right)}\;\left(\mathrm{GJ}\right)$	$\Delta h_{O}^{\left(x_{s}\right)}\;\left(\mathrm{GJ}\right)$	RPD%
2.6	0.3367	0.3325	1.2552	0.3328	0.3288	1.2092
5.2	0.5185	0.5137	0.9300	0.5157	0.5109	0.9351
7.8	0.6623	0.6572	0.7730	0.6607	0.6556	0.7749
10.4	0.7831	0.7780	0.6533	0.7828	0.7777	0.6536
13.0	0.8868	0.8823	0.5087	0.8882	0.8836	0.5192
	**FEM**			**RHBIM**		
**Days**	$\Delta h_{N}^{\left(x_{\ell }\right)}\;\left(\mathrm{GJ}\right)$	$\Delta h_{O}^{\left(x_{\ell }\right)}\;\left(\mathrm{GJ}\right)$	RPD%	$\Delta h_{N}^{\left(x_{\ell }\right)}\;\left(\mathrm{GJ}\right)$	$\Delta h_{O}^{\left(x_{\ell }\right)}\;\left(\mathrm{GJ}\right)$	RPD%
2.54	0.3298	0.3383	2.5445	0.3267	0.3350	2.5087
5.08	0.5084	0.5197	2.1982	0.5062	0.5173	2.1690
7.62	0.6492	0.6640	2.2540	0.6481	0.6628	2.2427
10.16	0.7667	0.7874	2.6639	0.7668	0.7875	2.6636
12.70	0.8666	0.8970	3.4475	0.8682	0.8988	3.4635

**Table 6 molecules-26-00365-t006:** RPD between both models and according to the numerical and semi-analytical solutions to the released thermal energy during a solidification process of the ${\mathrm{KNO}}_{3}/{\mathrm{NaNO}}_{3}$ salt.

	FEM			RHBIM		
**Days**	ΔhN(xs)(GJ)	ΔhO(xs)(GJ)	RPD%	ΔhN(xs)(GJ)	ΔhO(xs)(GJ)	RPD%
3.90	0.3675	0.3651	0.6552	0.3624	0.3601	0.6366
7.80	0.5893	0.5860	0.5615	0.5858	0.5826	0.5478
11.70	0.7628	0.7580	0.6312	0.7606	0.7559	0.6198
15.60	0.9021	0.8948	0.8125	0.9075	0.8938	1.5211
19.50	1.0154	1.0019	1.3384	1.0155	1.0020	1.3382
	**FEM**			**RHBIM**		
**Days**	ΔhN(xℓ)(GJ)	ΔhO(xℓ)(GJ)	RPD%	ΔhN(xℓ)(GJ)	ΔhO(xℓ)(GJ)	RPD%
3.98	0.3750	0.3754	0.1066	0.3750	0.3746	0.1067
7.96	0.5993	0.6043	0.8308	0.5963	0.6011	0.8017
11.94	0.7734	0.7789	0.7086	0.7717	0.7771	0.6973
15.92	0.9119	0.9174	0.6013	0.9113	0.9168	0.6017
19.90	1.0221	1.0268	0.4587	1.0227	1.0275	0.4682

## Data Availability

The data presented in this study are available on request from the corresponding author.

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
