# Peer review of "Effects of Total Thermal Balance on the Thermal Energy Absorbed or Released by a High-Temperature Phase Change Material"

_molecules, 2021, doi:10.3390/molecules26020365_

Round 1

Reviewer 1 Report

The comments are attached.

Reviewer 2 Report

This is the article on Thermal balance for Phase Change Material

I think the content is described reasonably well but needs some  revisions to address the following points before considering publication in this Journal.

  1. The abstract need to be improved in line with the introduction referring the example study of KNO3.
  2. Need to modify the word apparent latent heat in the abstract.
  3. The authors referred "Isobaric Phase Transition" for one dimensional Material and never referred afterwards neither the definition and nor an explanation.  Is KNO3 treated as one dimensional material?
  4. In the method and material the one dimensionality is not defined. 
  5. It is mentioned that the heat transfer through the system is homogeneous about the perpendicular plane.  I will love to see the directional impact in the mathematical form and reason for the homogeneity in one direction.
  6. As temperature is key , I wish to see an explicit definition of Tl, Ts and Tm.
  7. The authors mentioned: is the initial temperature distribution in the liquid (solid) domain, and T` (Ts) is the temperature distribution at some time value t.  But the equation is part of Finite Element approach.
  8. What is the model used in Finite Element method, can the author elaborate about the mesh size, the interface design, if it is a singe particle or multi-particle method. I as well wanted to know if the system is for single or multi-material .
  9. I want to see a comparison with other Finite element method result with the current one. 

Round 2

Reviewer 2 Report

I am satisfied with the response the authors provided and i therefore recommend publication with minor spell check.